# Effect of Jakyakgamcho-Tang Extracts on H_2_O_2_-Induced C2C12 Myoblasts

**DOI:** 10.3390/molecules26010215

**Published:** 2021-01-04

**Authors:** Young Sook Kim, Heung Joo Yuk, Dong Seon Kim

**Affiliations:** 1Research Infrastructure Team, Herbal Medicine Research Division, Korea Institute of Oriental Medicine, Daejeon 34054, Korea; ykim@kiom.re.kr; 2Herbal Medicine Research Division, Korea Institute of Oriental Medicine, Daejeon 34054, Korea; yukhj@kiom.re.kr

**Keywords:** antioxidant, C2C12 cell, Jakyakgamcho-tang, muscle atrophy, oxidative stress

## Abstract

Oxidative stress is a major contributor to muscle aging and loss of muscle tissue. Jakyakgamcho-tang (JGT) has been used in traditional Eastern medicine to treat muscle pain. Here, we compared the total phenolic and flavonoid contents in 30% ethanol and water extracts of JGT and tested the preventive effects against oxidative stress (hydrogen peroxide)-induced cell death in murine C2C12 skeletal muscle cells. The total phenolic content and total flavonoid content in 30% ethanol extracts of JGT were higher than those of water extracts of JGT. Ethanol extracts of JGT (JGT-E) had stronger antioxidant activities of 2,2′-azino-bis-3-ethylbenzothiazoline-6-sulfonic acid (ABTS) and 2,2′-diphenyl-1-picrylhydrazyl-scavenging activity (DPPH) than water extracts of JGT (JGT-W). JGT-E contained 19–53% (1.8 to 4.9-fold) more active compounds (i.e., albiflorin, liquiritin, pentagalloylglucose, isoliquiritin apioside, isoliquiritin, liquiritigenin, and glycyrrhizin) than JGT-W. The ethanol extracts of JGT inhibited hydrogen peroxide-induced cell death and intracellular reactive oxygen species generation more effectively than the water extract of JGT in a dose-dependent manner. For the first time, these results suggest that ethanol extract of JGT is relatively more efficacious at protecting against oxidative stress-induced muscle cell death.

## 1. Introduction

Sarcopenia, or aging-associated muscle loss, affects 10% of all adults aged over 50 years old. Muscle aging is characterized by a decline in function and a loss of tissue. Oxidative stress and chronic inflammation are the main mechanisms of skeletal muscle senescence and are associated with increased protein breakdown, decreased protein synthesis, mitochondrial dysfunction, and apoptosis [1,2]. Oxidative stress induces muscle aging. Reactive oxygen species (ROS) are naturally and constantly formed during normal cellular activity and act as mediator of cellular dysfunctions under condition of hypoxia, inflammation or in the impaired antioxidant defenses in vivo [3]. The increase in ROS levels has harmful effects on cell homeostasis, structure, and functions, and results in oxidative stress. Hydrogen peroxide (H_2_O_2_) induces atrophy and loss in myotubes, and it is used as an exogenous oxidant treatment to promote free radical generation in both isolated skeletal muscle and in in vitro myotubes [3].

Herbs are sources of important remedies and supplements used in the traditional medicine of Eastern Asia. Nearly 65% of the global population uses herbs as part of their primary health care modality [4]. About 25% of all modern medicines such as aspirin and ephedrine are derived from plants [5]. Natural antioxidants occur in herbal medicines and foods. Several studies have assessed the impacts of supplement and nutrients on muscle strength, mass, and physical performance [6]. It was found that dietary antioxidants such as vitamins C and E inhibited oxidation, induced antioxidant enzymes, and improved positive work function in the chronically loaded muscles of aged rats [7]. Vitamin C regenerates vitamin E in cell membranes, and the latter inhibits free radical production [6]. The study in [8] evaluated the free radical-scavenging abilities of natural products via in vitro assays, specifically their scavenging activities against 2,2′-azino-bis-3-ethylbenzothiazoline-6-sulfonic acid (ABTS), 2,2-diphenyl-1-picrylhydrazyl (DPPH), superoxide, and hydroxyl and nitric oxide radicals as well as their influences on lipid peroxidation levels and antioxidant enzyme activation. The oxidation of ABTS generates preformed radical mononcation of ABTS (ABTS·^+^) with potassium persulfate, and it reduces in the presence of hydrogen-donating antioxidants [8].

Jakyakgamcho-tang (JGT; Shaoyao-gancao-tang in Chinese; Shakuyaku-kanzo-to in Japanese) is a herbal medicine that is widely prescribed in traditional Eastern medicine and is used to alleviate muscle pain. JGT consists of *Paeonia lactiflora* Pallas radix (root) and *Glycyrrhiza uralensis* Fisch radix and rhizome. In a clinical trial, JGT reduced muscle pain and improved muscle damage after exercise [9]. JGT and its major component *Paeonia lactiflora* have potent antiglycation properties [10]. Glycation is a spontaneous nonenzymatic reaction of free reducing sugars and amino groups with protein and lipids in blood and tissues. Glycation is highly accelerated under oxidative stress and is implicated in the pathogenesis of diabetes and age-related diseases.

More recently, ethanol or ethanol with water has been used as an extraction solvent for traditional Chinese or herbal medicines for pharmaceuticals and dietary supplements [9,10]. However, to the best of our knowledge, no studies have assessed the efficacy of 30% ethanol extract of JGT (JGT-E) against aging-related muscular atrophy in vitro. Thus, we aim to analyze and evaluate the antioxidant properties of water extracts of JGT (JGT-W) and JGT-E. In addition, we evaluate the effect of JGT-E and JGT-W on H_2_O_2_-induced cell death in murine C2C12 skeletal muscle cells.

## 2. Materials and Methods

### 2.1. Preparation of 30% Ethanol and Water Extracts of Jakyakgamcho-Tang

*Paeonia lactiflora* and *Glycyrrhiza uralensis* roots were purchased from an oriental herbal market (Omniherb, GyeongsangBuk-Do, Korea) (http://www.omniherb.com/) that only handles herbs certified by the Korean Pharmacopoeia. The Jakyakgamcho-tang was prepared using a 2:1 (*w/w*) ratio of *P. lactiflora* root to *G. uralensis* root. The JGT extracts were prepared by accurately weighing out 15 g of *P. lactiflora* root and 7.5 g of *G. uralensis* root and by mixing them according to the Oriental Medicine Advanced Searching Integrated System (http://oasis.kiom.re.kr) of the Korea Institute of Oriental Medicine. Distilled water or 30% (*v/v*) ethanol was added to the roots, and the mixture was extracted for 1 h in a reflux extractor (MS-DM607; M-TOPS, Seoul, Korea). The extracts were then filtered, evaporated under reduced pressure in a rotary evaporator (N-1200A; Eyela, Tokyo, Japan) at 50 °C, and freeze-dried (FDU-2100; Eyela, Tokyo, Japan) at −80 °C for 72 h to obtain either a 30% (*v/v*) ethanol extract (JGT-E; yield 29.6%) or water extract (JGT-W; yield 34.8%). For ultra-performance liquid chromatography (UPLC), 20 mg of each extract was dissolved in 2 mL dimethyl sulfoxide (DMSO) and diluted tenfold with distilled water. The solutions were passed through a 0.45-μm syringe filter (Whatman; Clifton, NJ, USA) before injection.

### 2.2. UPLC-Quadruple Time-of-Flight Mass Spectrometry (UPLC-Q-Tof/MS) Analysis

UPLC was performed in a Waters Acquity UPLC system (Waters Corp., Milford, MA, USA) coupled to a Q-Tof Premier ESI/mass spectrometer (Q-Tof Premier™, Waters Corp., Milford, MA, USA). Three-microliter aliquots per sample were injected into a BEH C18 column (100 mm × 2.1 mm; i.d. 1.7 μm) at a flow rate of 0.4 mL min^−1^ and eluted with a chromatographic gradient, comprising mobile phases A (water containing 0.1% (*v/v*) formic acid) and B (acetonitrile containing 0.1% (*v/v*) formic acid). The linear gradient was optimized as follows: 0 min, 10% B; 0–9 min, 10–20% B; 9–11 min, 20–30% B; 11–12 min, 30–90% B; 12–13 min, 90–100% B; 13–14 min, 100% B; and 14–15 min, 10% B. The Q-Tof was operated in negative ion mode under the following conditions: capillary voltage, 2.3 kV; cone voltage, 50 V; source temperature, 110 °C; and desolvation temperature, 350 °C. A sprayer with a leucine-enkephalin reference solution ([M−H] − m/z 554.2615) served as the lock mass. Full scan data and MS/MS spectra were collected with MassLynx 4.1 software (Waters Corp., Milford, MA, USA).

### 2.3. Determination of Total Phenolic Content (TPC) and Total Flavonoid Content (TFC) in JGT-W and JGT-E

The total phenolic content (TPC) in various extracts of JGT and its constituent herbs were determined by a modified Folin–Ciocalteu method [11]. Briefly, 0.125 mL of each extract was transferred to a test tube containing 1.8 mL Folin–Ciocalteu reagent. After 5 min, 1.2 mL of 15% (*w/v*) Na_2_CO_3_ was added, and the mixture was kept in the dark at 25 °C for 2 h. Absorbance was measured at 765 nm in a SpectraMax M2 multimode microplate reader (Molecular Devices, Sunnyvale, CA, USA). A linear calibration curve (R^2^ = 0.998) was plotted using gallic acid, and the results were expressed in mg gallic acid equivalents (GAE) per g extract sample.

Total flavonoid content (TFC) was determined by an aluminum chloride colorimetric assay [11]. Quantification was expressed by reporting the absorbance in the quercetin calibration graph used as the flavonoid standard (R^2^ = 0.999). A 1 mL sample of prepared extract or standard was mixed with 4 mL distilled water and 0.3 mL of 5% (*w/v*) NaNO_2_. After 5 min and 1 min, respectively, 0.3 mL of 10% (*w/v*) AlCl_3_ and 2 mL of 1 M NaOH were added. Then, 2.4 mL distilled water was added, and the mixture was shaken. The resultant pink color was read in a SpectraMax M2 multimode microplate reader at 510 nm, and the results were expressed in mg quercetin equivalents (QE) per g extract sample.

### 2.4. Scavenging of 2,2-Azino-Bis-3-Ethylbenzothiazoline-6-Sulfonic Acid (ABTS)

ABTS-scavenging activity was assessed according to a modified version of the method described by Re et al. [12]. ABTS (7.0 mM) and potassium persulfate (2.45 mM) were mixed in water and stored at room temperature for 12 h in the dark to produce ABTS·+. The aqueous ABTS·+ solution was then diluted to an absorbance of 1.50 at 405 nm. Various concentrations of each extract solution (1 mL) were added to 2 mL diluted ABTS·+ solution, and the absorbance were measured at 405 nm.

### 2.5. 2,2′-Diphenyl-1-Picrylhydrazyl (DPPH) Free Radical Assay

Water and 30% (*v/v*) ethanol extracts of the *P. lactiflora* and *G. uralensis* roots (50 μg mL^−1^, 100 μg mL^−1^, 250 μg mL^−1^, 500 μg mL^−1^, and 1000 μg mL^−1^) were prepared for the DPPH assay. The DPPH-scavenging activity of the extracts was measured according to a previously reported method [8]. Here, 1 mm aliquots of the various concentrations of the extract solutions were added to 2 mm of a DPPH-methanol solution (5 mg 100 mL^−1^). The decrease in absorbance at 517 nm was measured with a SpectraMax M2 multimode microplate reader. The DPPH-scavenging activity (%) was calculated according to the following equation:(1)$$\mathrm{DPPH}\;\mathrm{scavenging}\;\mathrm{activity}\;(\%)\;=\;\left[1\;-\;\frac{A\;sample}{A0}\right]\;\times \;100$$
where A*_sample_* is the steady-state absorbance of the sample solution, and A_0_ is the absorbance of the DPPH solution before the addition of the extract.

### 2.6. Cell Culture

The immortalized mouse myoblast cell line C2C12 was purchased from the American Type Culture Collection (ATCC CRL-1772; Manassas, VA, USA) and cultured in the Dulbecco’s Modified Eagle Medium (DMEM), supplemented with 10% (*v/v*) fetal bovine serum (FBS) and 1% (*w/v*) penicillin-streptomycin (P/S) at 37 °C in a humidified 5% CO_2_ incubator as per previously described methods [13]. To induce myotube formation, the media were replaced with differentiation media (DMEM with 2% (*v/v*) FBS and 1% (*w/v*) P/S) and cultured for an additional 7 d.

### 2.7. Cell Viability Assay in H_2_O_2_-Induced C2C12 Cells

C2C12 viability was determined by a cell counting kit-8 (CCK-8) assay (Dojindo Molecular Technologies, Kumamoto, Japan). C2C12 cells (0.5 × 10^4^ cells/well) were seeded in each well of a 96-well plate containing DMEM plus 10% (*v/v*) FBS and incubated for 24 h. After attachment, the cells were pretreated with JGT extract in a serum-free medium for 24 h and then with 250 μM H_2_O_2_ for 3 h. After incubation, 10 μL CCK-8 was added to each well, and the plate was incubated at 37 °C and 5% CO_2_ in a humidified incubator for 2 h. Absorbance was measured at 450 nm in a multidetection microplate reader (Synergy HT; BioTek, Winooski, VT, USA).

### 2.8. Intracellular ROS Measurements in H_2_O_2_-Induced C2C12 Cells

C2C12 cells were pretreated with JGT-E and JGT-W for 24 h and then subjected to 250 μM H_2_O_2_ for 30 min. *N*-acetylcysteine (NAC, 1 mM) served as a positive control. Intracellular ROS levels were measured by the DCF-DA method, wherein the fluorescent probe 2′,7′-dichlorodihydrofluorescein diacetate (H_2_DCF-DA, Molecular Probes Inc., Eugene, OR, USA) is converted by intracellular esterases to 2′,7′-dichlorodihydrofluorescein, which in turn is oxidized by intracellular ROS to highly fluorescent 2′,7′-dichlorofluorescein (DCF). The treated cells were then washed with a Hank’s Balanced Salt Solution (HBSS) buffer and incubated in the dark for 30 min in a HBSS buffer containing 50 μM H_2_DCF-DA. DCF fluorescence was measured in a Synergy HT spectrofluorometer (BioTek, Winooski, VT, USA) at an excitation wavelength of 485 nm and an emission wavelength of 535 nm. All experiments were repeated at least thrice.

### 2.9. Statistical Analysis

Data are means ± SD or mean ± SE. Here, ANOVA with Tukey’s test was used for multiple treatment means comparisons in Prism v. 7.0 (GraphPad Software, San Diego, CA, USA).

## 3. Results

### 3.1. Qualitative and Quantitative Analyses of JGT Components by UPLC-PDA-ESI-QTOF-MS

As shown in Figure 1 and Figure 2, the major active compounds in JGT-E and JGT-W were albiflorin (1), paeoniflorin (2), liquiritin (3), pentagalloylglucose (4), isoliquiritin apioside (5), isoliquiritin (6), liquiritigenin (7), and glycyrrhizin (8). JGT-E contained 19–53% more of the aforementioned constituents than those in JGT-W, except for albiflorin (1) (Figure 3).

### 3.2. Effects of JGT-E and JGT-W on ABTS- and DPPH-Scavenging Activities

The relative total phenolic content (TPC) and total flavonoid content (TFC) are shown in Table 1. The TPC and TFC in the 30% (*w/v*) ethanol extracts of *P. lactiflora* and *G. uralensis* roots were higher than those in the water extracts. The contents of TPC and TFC were higher than those in JGT-W.

Table 2 shows the antioxidant properties of water and 30% (*w/v*) ethanol extracts evaluated by in vitro ABTS+ − and DPPH-scavenging assays. The antioxidant activity of JGT-E was higher than that of JGT-W. The ABTS+− and DPPH-scavenging capacities JGT-E are commensurate with the phenolic compound content.

### 3.3. JGT-E and JGT-W Protect C2C12 Myocytes against H_2_O_2_-Induced Decrease in Cell Viability

To determine whether hydrogen peroxide (H_2_O_2_) induced C2C12 death, we subjected the cells to various H_2_O_2_ concentrations for 24 h (Figure 4A). H_2_O_2_ decreased C2C12 viability in a dose-dependent manner. To assess the efficacies of JGT-E and JGT-W at preventing H_2_O_2_-induced C2C12 death, we pretreated the cells with JGT-E and JGT-W, subjected them to H_2_O_2_, and evaluated cell viability by CCK-8. Figure 4B shows that JGT-E and JGT-W preserve C2C12 proliferative activity in a dose-dependent manner.

### 3.4. JGT-E and JGT-W Protect C2C12 Myocytes from H_2_O_2_-Induced Damage

The H_2_O_2_ treatment was cytotoxic to C2C12, but both JGT-W and JGT-E helped prevent cell death. To determine whether this protection was related to the antioxidant effects of JGT-W and JGT-E in C2C12, we evaluated whether JGT-W and JGT-E could inhibit intracellular ROS generation in H_2_O_2_-treated cells. Figure 5 shows that the nearly fourfold increase in intracellular ROS levels induced by H_2_O_2_ was significantly inhibited by JGT-W and JGT-E pretreatment in a dose-dependent manner. JGT-E reduced DCF fluorescence by >27.75% (100 μg mL^−1^) and by 25.50% (50 μg mL^−1^) compared to JGT-W. NAC pretreatment also attenuated H_2_O_2_-stimulated increase in intracellular ROS levels in C2C12.

## 4. Discussion

During aging, oxidative stress plays an important role in the progression of muscle mass loss or skeletal muscle atrophy [14]. Dietary supplementation with exogenous antioxidants such as herbal medicines exerts health-promoting effects and helps to prevent muscle aging [15]. Jakyakgamcho-tang (JGT), a herbal medicine consisting of a 2:1 (*w/w*) mixture of *Paeonia lactiflora* and *Glycyrrhiza uralensis* roots, has been used both clinically and pharmacologically in the treatment of muscle and acute abdominal pain and backache [16,17]. Here, we analyzed the composition of a water extract (JGT-W) and a 30% (*w/v*) ethanol extract of JGT (JGT-E) and confirmed whether the efficacies of JGT-W and JGT-E could prevent muscle cell damage under oxidative stress for the first time.

JGT is a herbal medicine widely prescribed in Eastern Asia and has recently been known for various effects, such as the inhibitory effect of formation of advanced glycation end products, the inhibitory effect of nitric oxide, and anti-inflammatory effect [9,10]. Here, Figure 3 showed the antioxidant properties of JGT-W and JGT-E by in vitro ABTS- and DPPH-scavenging assay, and the antioxidant activity of JGE-E was higher than that of JGT-W. In addition, JGT-E had higher total phenolic and flavonoid content than those in JGT-W (Table 1). Plant phenolic and flavonoid compounds are secondary metabolites, and these compounds scavenge free radicals and act as antioxidants [11,18,19]. JGT-E was relatively more effective at scavenging ABTS·+ and DPPH-inhibiting intracellular ROS generation in skeletal muscle cells as it contained comparatively more TPC and TFC. A previous study showed that phenolic compounds strongly inhibit H_2_O_2_-induced apoptosis in HepG2 cells [20]. To date, no studies have investigated the contents of plant phenolic and flavonoid compounds in JGT-W and JGT-E.

One previous study showed that the relative amounts of effective components such as paeoniflorin, benzoic acid, glycyrrhizin, and isoliquiritin in JGT is higher in the 70% ethanol extracts than in water-extracted JGT [21]. Given the herbal combination of JGT and with chromatographic conditions, various compounds in JGT can be isolated [22]. As shown as Figure 3 and Table 1, the levels of all measured phenolic compounds except albiflorin were also higher in JGT-E than JGT-W. Paeoniflorin was found to be the most abundant component, and its concentration markedly differed between JGT-E and JGT-W. This compound is a pinnae monoterpene glycoside with various pharmacological effects, such as antioxidant, anti-inflammatory, and immunoregulatory effects [23,24]. Liquiritin has an antioxidant effect on neuroblastoma cells [25,26]. Pentagalloylglucose prevents acute lung injury in lipopolysaccharide-induced rat models and has antioxidant properties [27,28,29,30,31]. Isoliquiritin apioside shows marked modulatory activities against oxidative stress-induced genotoxicity [31]. Isoliquiritin presents with antioxidant activity, while liquiritigenin is also a potent antioxidant and shows various efficacy against multidrug-resistant bacteria [32,33]. Glycyrrhizin reduces the formation of ROS by inducing AMPK phosphorylation and nuclear transfer NRF2, resulting in an upregulation of antioxidant enzymes and anti-inflammatory efficacy [34,35].

The accumulation of oxidative stress leads to cell transformation and causes tissue injury, loss of function, enhanced senescence, and loss of potential associated with degenerative diseases associated with aging such as sarcopenia [36]. Excess cellular levels of ROS cause damage to cell membrane and organelles, which leads to activation of cell death [37]. Here, ROS-induced cell death was prevented in JGT-E or JGT-W-pretreated C2C12 cells via the inhibition of oxidation (Figure 4B). The inhibition of oxidative stress by JGT-E is thought to contribute to promoting cell growth. Exogenous antioxidants that reduce intracellular ROS levels promote cell growth in tumor progression [38].

Generally, traditional Chinese and oriental herbal medicines are water extracts. The Korea Food and Drug Administration either exempts or requires minimum toxicity test data for the approval of oriental herbal medicines when the ethanol content in the extraction solvent is ≤30% and the balance is water. The present study demonstrated that an ethanol-water solvent mixture was relatively more effective at extracting the bioactive components in JGT, thereby increasing its biological activity and reducing the dosage required for optimal efficacy. JGT-E showed the preventive effect on H_2_O_2_-induced cell death via the inhibition of oxidation. To the best of our knowledge, this is the first report on the effectiveness of JGT-E against aging-related muscular atrophy in vitro.

## 5. Conclusions

The aim of this study was to compare the relative antioxidant principles of JGT-W and JGT-E derived from the traditional herbal blend, JGT. JGT-E had strong antioxidant activity and prevented oxidative stress-induced skeletal muscle cell death. The results of the present study suggest that low doses of JGT-E may help prevent skeletal muscle aging via the inhibition of oxidation. Moreover, 30% (*v/v*) ethanol effectively extracts JCT and substantially enhances its antioxidant and other biological activities compared with traditional water extracts that have been prepared to reduce the dosages for muscle atrophy.

## Figures and Tables

**Figure 1 molecules-26-00215-f001:**
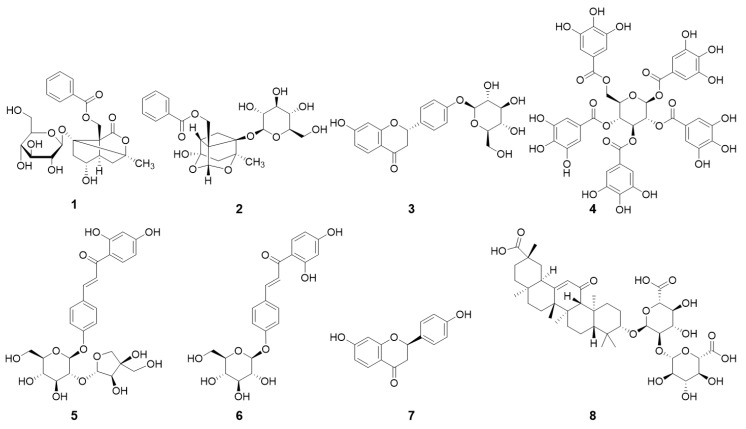
Chemical structures of Jakyakgamcho-tang (JGT) components: albiflorin (1), paeoniflorin (2), liquiritin (3), pentagalloylglucose (4), isoliquiritin apioside (5), isoliquiritin (6), liquiritigenin (7), and glycyrrhizin (8).

**Figure 2 molecules-26-00215-f002:**
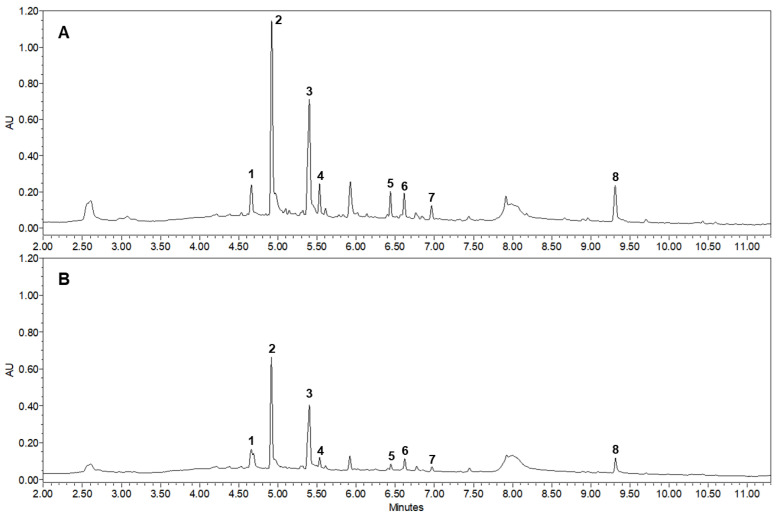
Ultra-performance liquid chromatography profiles for JGT-E (ethanol extracts of JGT) and JGT-W (water extracts of JGT). (**A**) JGT-E, 30% (*w/v*) EtOH extract of Jakyakgamcho-tang; (**B**) JGT-W, water extract of Jakyakgamcho-tang. Albiflorin (1), paeoniflorin (2), liquiritin (3), pentagalloylglucose (4), isoliquiritin apioside (5), isoliquiritin (6), liquiritigenin (7), and glycyrrhizin (8).

**Figure 3 molecules-26-00215-f003:**
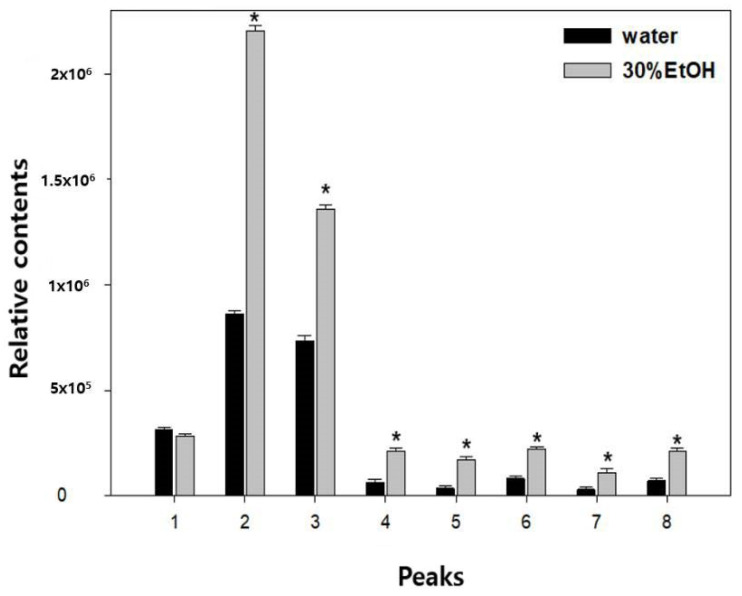
Comparative changes in individual peak marker levels from two extraction solvents. Calculations were based on the area determined from UPLC spectra. Mean ± SD of measurements determined for three independent samples analyzed thrice (*n* = 3). * *p* < 0.001 vs. water extract. Albiflorin (1), paeoniflorin (2), liquiritin (3), pentagalloylglucose (4), isoliquiritin apioside (5), isoliquiritin (6), liquiritigenin (7), and glycyrrhizin (8).

**Figure 4 molecules-26-00215-f004:**
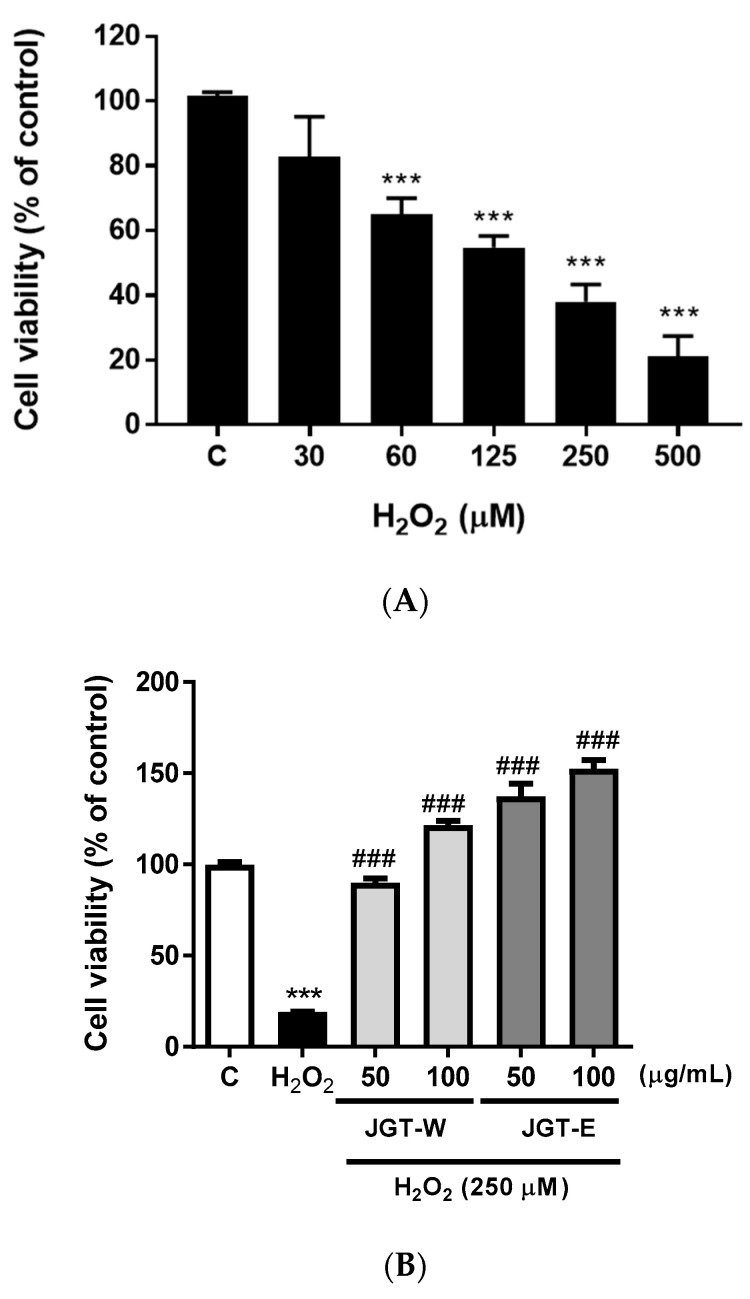
Effects of JGT-W and JGT-E on growth of H_2_O_2_-treated C2C12 cells. (**A**) H_2_O_2_-induced C2C12 death. (**B**) Protective effects of JGT-W and JGT-E in C2C12 against H_2_O_2_-induced cytotoxicity. Data are representative of three independent experiments and expressed as mean ± standard error of mean (*n* = 7). *** *p* < 0.001 vs. control; ^###^
*p* < 0.001 vs. viability of H_2_O_2_-treated cells. JGT-E, 30% (*w/v*) ethanol extract of Jakyakgamcho-tang, JGT-W, Jakyakgamcho-tang water extract.

**Figure 5 molecules-26-00215-f005:**
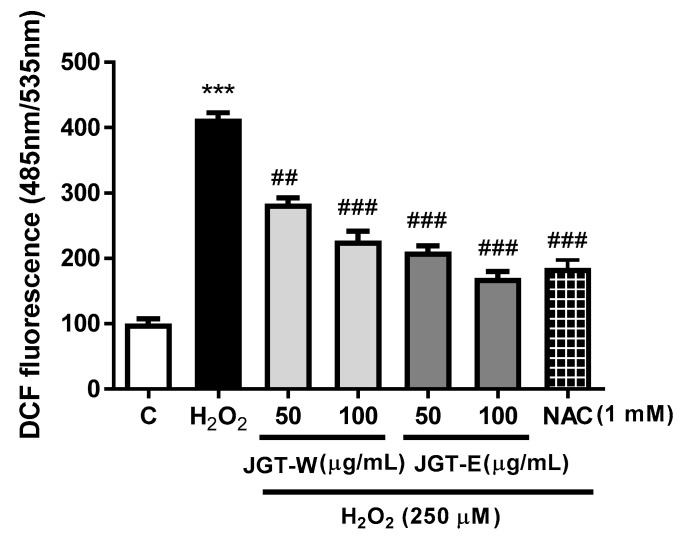
Effects of JGT-W and JGT-E extracts on viability of H_2_O_2_-treated C2C12 cells. NAC was positive control. Data are representative of three independent experiments and expressed as mean ± standard error of mean (*n* = 7). *** *p* < 0.001 vs. control; ^##^
*p* < 0.01, ^###^
*p* < 0.001 vs. viability of H_2_O_2_-treated cells. JGT-E, 30% (*w/v*) ethanol extract of Jakyakgamcho-tang; JGT-W, Jakyakgamcho-tang water extract.

**Table 1 molecules-26-00215-t001:** Total phenolic content (TPC) and total flavonoid content (TFC) in Jakyakgamcho-tang (JGT) and its constituents.

Sample	TPC (mg GAE g^−1^)	TFC (mg QE g^−1^)
Water extract of *P. lactiflora* root	17.17 ± 0.6	3.25 ± 0.3
30% (*w/v*) ethanol extract of *P. lactiflora* root	29.70 ± 1.0 *	8.11 ± 0.1 *
Water extract of *G. uralensis* root	18.37 ± 0.1	3.72 ± 0.1
30% (*w/v*) ethanol extract of *G. uralensis* root	23.10 ± 1.6 *	7.02 ± 0.2 *
JGT-W	25.49 ± 0.8	3.94 ± 0.5
JGT-E	34.01 ± 2.6 **	7.32 ± 0.6 **

Data indicate mean ± SD (*n* = 3). All extracts were analyzed in triplicate. * *p* < 0.05 vs. water extract; ** *p* < 0.01 vs. JGT-W.

**Table 2 molecules-26-00215-t002:** Relative ABTS and DPPH activities of Jakyakgamcho-tang (JGT).

Samples	ABTS		DPPH	
	Inhibition (%)	IC_50_ (µg mL^−1^)	Inhibition (%)	IC_50_ (µg mL^−1^)
JGT-W	72.8 ± 1.2	60.5 ± 0.9	25.1 ± 1.3	742.8 ± 2.1
JGT-E	80.8 ± 1.1	34.2 ± 0.4 ***	38.9 ± 1.1	436.1 ± 3.7 ***

Data indicate mean ± SD (*n* = 3). All extracts were analyzed in triplicate. *** *p* < 0.0001 vs. JGT-W. ABTS, 2,2′-azino-bis-3-ethylbenzothiazoline-6-sulfonic acid; DPPH, 2,2-diphenyl-1-picrylhydrazyl; IC_50_, 50% inhibitory concentration; JGT-E, 30% (*w/v*) ethanol extract of Jakyakgamcho-tang, JGT-W, Jakyakgamcho-tang water extract.

## Data Availability

All the data generated and analyzed in this study are mentioned in this manuscript.

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
