# Peer review of "Effect of Jakyakgamcho-Tang Extracts on H2O2-Induced C2C12 Myoblasts"

_molecules, 2021, doi:10.3390/molecules26010215_

Round 1
Reviewer 1 Report
The present paper aims to show differences between JGT water and ethanolic extracts. The Authors can find below some comments and suggestions to make the manuscript suitable to be published in Molecules
Abstract
It does not make sense to have put number labels for each compound in the Abstract section (Line 18-20). They should be removed
Introduction
Line 45-46: I have not grasped the sense of quoting vitamin C functions and vitamin E, since they are not present in JGT extracts.
Material and methods
Line 137 2.5. is actually 2.7.
Line 145 2.6 is actually 2.8
Line 159 2.7 is actually 2.9
Results
Line 203-204: "Figure 4B shows that JGT-E and JGT-W preserve C2C12 viability in a dose-dependent manner" is not correct. The graph shows that cell growth has increased with respect to control cells and in spite of the presence of H2O2. At this point it would be interesting to see the effect of single compounds without H2O2 treatments, just to confirm the proliferative activity induced by JGT extracts.
Discussion
In my opinion the Discussion must be improved, since it seems merely a list with unrelated phrases (for instance, lines 248-257 or 260-266). It would also be useful to better discuss the results and explain what could have happened by comparing the outcomes with what can be found in literature.
Line 242: Any comment about the extraction capacity of ethanol vs water, which determined higher content in polyphenols and antioxidant activity? After all, the difference between the two extracts is based on this.
Line 261 - Do the Authors have any explanation for the increase of cell growth? It should be properly discussed
Author Response
Ref.: Ms. ID. Molecules-1032714
Thank you for email on December 3, 2020, informing us of your comments on our manuscript (Manuscript ID: molecules-1032714, Title: Effect of Jakyakgamcho-tang Extracts on H2O2-Induced C2C12 Myoblasts). I would now like to resubmit the revised manuscript. The manuscript has been modified based on your comments and corrections, our replies to which are included below. In the manuscript, we mark the highlight in the changed part.
Response to Reviewer 1 Comments
Comments and Suggestions for Authors
The present paper aims to show differences between JGT water and ethanolic extracts. The Authors can find below some comments and suggestions to make the manuscript suitable to be published in Molecules
Abstract
It does not make sense to have put number labels for each compound in the Abstract section (Line 18-20). They should be removed
- Numbers were removed
Introduction
Line 45-46: I have not grasped the sense of quoting vitamin C functions and vitamin E, since they are not present in JGT extracts.
- As reviewer’s comments, vitamin C and E are not present in JGT extracts However in this point JGT extracts showed the anti-oxidant effect such as vitamin C and E.
Material and methods
Line 137 2.5. is actually 2.7. => corrected
Line 145 2.6 is actually 2.8 => corrected
Line 159 2.7 is actually 2.9 => corrected
Results
Line 203-204: "Figure 4B shows that JGT-E and JGT-W preserve C2C12 viability in a dose-dependent manner" is not correct. The graph shows that cell growth has increased with respect to control cells and in spite of the presence of H2O2. At this point it would be interesting to see the effect of single compounds without H2O2 treatments, just to confirm the proliferative activity induced by JGT extracts.
- Line 204: Viability was changed proliferative activity. JGT treatment without H2O2 increased but not significant effect on the viability of C2C12, implying that JGT is nontoxic in the cells (Data not shown). It is estimated that the preventive effect of H2O2-induced cell damage in JGT treatment with H2O2 is greater than JGT treatment and its effect shows a significant increase in efficacy.
Discussion
In my opinion the Discussion must be improved, since it seems merely a list with unrelated phrases (for instance, lines 248-257 or 260-266). It would also be useful to better discuss the results and explain what could have happened by comparing the outcomes with what can be found in literature.
- As reviewer’s comments, that part was changed specially references explained as antioxidant effects.
Line 242: Any comment about the extraction capacity of ethanol vs water, which determined higher content in polyphenols and antioxidant activity? After all, the difference between the two extracts is based on this.
- Yes, the difference of the higher content in polyphenols and antioxidant activity is the extraction solvents.
Line 261 - Do the Authors have any explanation for the increase of cell growth? It should be properly discussed
- We added this sentence in the discussion part: Inhibition of oxidative stress by JGT-E is thought to contribute to promoting cell growth.
Thank you for your consideration. I look forward to hearing from you.
Sincerely,
Dong-Seon Kim Ph.D.
Research Infrastructure Team, Herbal Medicine Research Division,
Korea Institute of Oriental Medicine,
Postal address: 1672 Yuseongdae-ro, Yuseong-gu, Daejeon, 34054, Republic of Korea
Telephone: +82-42-868-9639 Fax: + 82-42-868-9578
E-mail: dskim@kiom.re.kr

Reviewer 2 Report
The author had correct the manuscript according to my comments.
Author Response
Ref.: Ms. ID. Molecules-1032714
Thank you for email on December 3, 2020, informing us of your comments on our manuscript (Manuscript ID: molecules-1032714, Title: Effect of Jakyakgamcho-tang Extracts on H2O2-Induced C2C12 Myoblasts). I would now like to resubmit the revised manuscript.
Response to Reviewer 2 Comments
Comments and Suggestions for Authors
The author had correct the manuscript according to my comments.
Thank you for your consideration. I look forward to hearing from you.
Sincerely,
Dong-Seon Kim Ph.D.
Research Infrastructure Team, Herbal Medicine Research Division,
Korea Institute of Oriental Medicine,
Postal address: 1672 Yuseongdae-ro, Yuseong-gu, Daejeon, 34054, Republic of Korea
Telephone: +82-42-868-9639 Fax: + 82-42-868-9578
E-mail: dskim@kiom.re.kr

Round 2
Reviewer 1 Report
The Authors have made some modifications as suggested, but the manuscript still need to be enhanced.
Discussion is still presented in a really rough form, full of sentences scattered here and there without a logical direction. It should be really improved to give depth to the outcome of the study.
Line 261-262 "Inhibition of oxidative stress by JGT-E is thought to contribute to promoting cell growth". Any references to justify this fact? In my opinion it should be better discussed, since it is really strange that control cells, which are supposed not to face oxidative stress, showed less proliferative activity than cell treated with JGT-E and with pro-oxidant H2O2.
Author Response
Ref.: Ms. ID. Molecules-1032714
Thank you for email on December 14, 2020, informing us of your comments on our manuscript (Manuscript ID: molecules-1032714, Title: Effect of Jakyakgamcho-tang Extracts on H2O2-Induced C2C12 Myoblasts). I would now like to resubmit the revised manuscript. The manuscript has been modified based on your comments and corrections, our replies to which are included below. In the manuscript, we mark the highlight in the changed part.
Response to Reviewer 1 Comments
Discussion is still presented in a really rough form, full of sentences scattered here and there without a logical direction. It should be really improved to give depth to the outcome of the study.
- Discussion was changed (page 8~9)
Line 261-262 "Inhibition of oxidative stress by JGT-E is thought to contribute to promoting cell growth". Any references to justify this fact? In my opinion it should be better discussed, since it is really strange that control cells, which are supposed not to face oxidative stress, showed less proliferative activity than cell treated with JGT-E and with pro-oxidant H2O2.
- Some papers published and cited that inhibition of oxidative stress promote cell growth in tumor cells. ROS production in normal cells is controlled by the appropriate regulation. Interestingly, ROS also is initiate cell growth in the progression of angiogenesis and metastasis. At moderate concentration, ROS activates the cancer survival signaling such as MAPK, p38, and pJNK. At high concentration, ROS can cancer cell apoptosis. In this experiment, control cells are controlled and cells treated with high concentration of H2O2 induced cell death.
- Although there was oxidative stress, the components of JGT are considered to inhibit oxidative stress and promote cell growth. Now we do not know which ingredients can inhibit oxidative stress and promote cell growth. More research is needs.
The present paper aims to show differences between JGT water and ethanolic extracts. The Authors can find below some comments and suggestions to make the manuscript suitable to be published in Molecules.
Thank you for your consideration. I look forward to hearing from you.
Sincerely,
Dong-Seon Kim Ph.D.
Research Infrastructure Team, Herbal Medicine Research Division,
Korea Institute of Oriental Medicine,
Postal address: 1672 Yuseongdae-ro, Yuseong-gu, Daejeon, 34054, Republic of Korea
Telephone: +82-42-868-9639 Fax: + 82-42-868-9578
E-mail: dskim@kiom.re.kr

Round 3
Reviewer 1 Report
The Authors have improved the overall quality of the manuscript, so now I recommend its publication on Molecules
Author Response
- According to review’s comments, we changed Figure 1.
This manuscript is a resubmission of an earlier submission. The following is a list of the peer review reports and author responses from that submission.
Round 1
Reviewer 1 Report
General comment:
The present paper aims to describe the potential antioxidant effects of two different extract coming from Jakyakgamcho-tang and tested on muscle cells. There is no noticeable advance from the analytical point of view, the research lacks innovation and no scientific novelty can be noticed. Overall, the manuscript is worthy of being published in Molecules, but only after substantial major revisions.
Other comments:
Abstract
Line 17: is ABTS, not ADTS
- No mention of the chemical characterization was made throughout the abstract. I think it should be included as it is an important part of the study.
Introduction
- The knowledge gap and the aim of the present work is not clearly mentioned. It should be better presented in the Introduction section.
Line 30-32: It is just a sequence of sentences without any causal link, it should be properly rewritten. Moreover, it is true that Hydroxyl radical (·OH), superoxide anion (O2-), and hydrogen peroxide (H2O2) cause oxidative damage to cells, but it is a matter of an abnormal production of ROS, not physiological as stated earlier. Please clarify this aspect.
Material and methods
Line 144: Hank’s Balanced Salt Solution must be written with the same font size of the manuscript
Results
A statistical comparison between JGT-W and JGT-E phytochemicals, total polyphenols content and antioxidants activity (Figure 3, Table 1 and Table 2) should be assessed to highlights significative differences to be reported in the Results section.
Line 190: Figure is 4B, not 3B
Figure 4 and 5: please pay attention to µ symbol, it can’t be seen clearly
Figure 4B: it is very strange that cell viability increases in muscle cells treated with extracts and H2O2 more than untreated cells (controls), it looks like there was an enhancement of cell proliferation induced by those substances. How do you explain it? Please discuss this fact in the Discussion section
Discussion
The Discussion section is very poorly drafted, it seems just a repetition of the Results with some other assertions here and there in the text. It must be properly redrafted, by discussing the main outcomes of the study and explaining the importance of the results.
Author Response
October 16, 2020
Molecules
Editor:
Ref.: Ms. ID. Molecules-960068
Thank you for email on October 9, 2020, informing us of your comments on our manuscript (Manuscript ID: molecules-960068, Title: Effect of Jakyakgamcho-tang Extracts on H2O2-Induced C2C12 Myoblasts). I would now like to submit the revised manuscript to be considered for publication as an articles in the molecules. The manuscript has been modified based on your comments and corrections, our replies to which are included below. In the manuscript, we mark the highlight in the changed part.
Response to Reviewer 1 Comments
General comment:
The present paper aims to describe the potential antioxidant effects of two different extract coming from Jakyakgamcho-tang and tested on muscle cells. There is no noticeable advance from the analytical point of view, the research lacks innovation and no scientific novelty can be noticed. Overall, the manuscript is worthy of being published in Molecules, but only after substantial major revisions.
Other comments:
Abstract
Line 17: is ABTS, not ADTS
- No mention of the chemical characterization was made throughout the abstract. I think it should be included as it is an important part of the study.
=> ADTS was changed to ABTS. The chemical characterization of ABTS was mentioned to introduction (line 46-48).
Introduction
- The knowledge gap and the aim of the present work is not clearly mentioned. It should be better presented in the Introduction section.
=> The water extract of JGT prescription is used to alleviate muscle pain not for muscle aging in Asian countries. One main mechanism of muscle aging is the oxidation that the oxidative stress induced muscle cell damage and mitochondrial dysfunctions. This paper was compared of water and 30% ethanol extracts in antioxidant properties and studied the preventive effect of oxidative stress-induced muscle cells death for the first time. The part of introduction was changed line 59~63.
Line 30-32: It is just a sequence of sentences without any causal link, it should be properly rewritten. Moreover, it is true that Hydroxyl radical (·OH), superoxide anion (O2-), and hydrogen peroxide (H2O2) cause oxidative damage to cells, but it is a matter of an abnormal production of ROS, not physiological as stated earlier. Please clarify this aspect.
=> The sentence was changed according to the reviewer’s comment (Line 31~36).
Material and methods
Line 144: Hank’s Balanced Salt Solution must be written with the same font size of the manuscript
=> Changed (line 148)
Results
A statistical comparison between JGT-W and JGT-E phytochemicals, total polyphenols content and antioxidants activity (Figure 3, Table 1 and Table 2) should be assessed to highlights significative differences to be reported in the Results section.
Line 190: Figure is 4B, not 3B
=> Changed
Figure 4 and 5: please pay attention to µ symbol, it can’t be seen clearly
=> Changed
Figure 4B: it is very strange that cell viability increases in muscle cells treated with extracts and H2O2 more than untreated cells (controls), it looks like there was an enhancement of cell proliferation induced by those substances. How do you explain it? Please discuss this fact in the Discussion section
=> It is not significant difference on cell viability increase the between the untreated cells (controls) and JGT-W (50 ug/ml). Low dose of JGT-W was co-treated with H2O2. This is the preventive effect low dose of JGT-W on H2O2-induced cell death.
Discussion
The Discussion section is very poorly drafted, it seems just a repetition of the Results with some other assertions here and there in the text. It must be properly redrafted, by discussing the main outcomes of the study and explaining the importance of the results.
=> Discussion section was rewritten.
I look forward to hearing from you concerning the acceptability of our manuscript in the future.
With best regards.
Sincerely,
Dong-Seon Kim, Ph.D.
Chief Research Scientist
Encl.
Co-authors

Reviewer 2 Report
This manuscript was discussed the antioxidant effect induce by H2O2 on C2C12 cell of JGT extracts. I have some major comments about this manuscript below:
1.The JGT extracts seems not novel for this filed of study, because many literature had discussed the antioxidant ability and functionality of JGT, Shaoyao-gancao-tang, Shakuyaku-kanzo-to. So the author should mention about the difference between the literature or novel findings in this manuscript.
2.In introduction, the author mention about the oxidation and sarcopenia, but in the conclusion, the mechanism of anti-sarcopenia for this materia was not showed.
Author Response
October 165, 2020
Molecules
Editor:
Ref.: Ms. ID. Molecules-960068
Thank you for email on October 9, 2020, informing us of your comments on our manuscript (Manuscript ID: molecules-960068, Title: Effect of Jakyakgamcho-tang Extracts on H2O2-Induced C2C12 Myoblasts). I would now like to submit the revised manuscript to be considered for publication as an articles in the molecules. The manuscript has been modified based on your comments and corrections, our replies to which are included below. In the manuscript, we mark the highlight in the changed part.
Response to Reviewer 2 Comments
This manuscript was discussed the antioxidant effect induce by H2O2 on C2C12 cell of JGT extracts. I have some major comments about this manuscript below:
1.The JGT extracts seems not novel for this field of study, because many literatures had discussed the antioxidant ability and functionality of JGT, Shaoyao-gancao-tang, Shakuyaku-kanzo-to. So the author should mention about the difference between the literature or novel findings in this manuscript.
=> Previous studies show that the water extract of JGT prescription is used to alleviate muscle pain not for muscle aging in Asian countries. One main mechanism of muscle aging is the oxidation that the oxidative stress induced muscle cell damage and mitochondrial dysfunctions. This paper was compared of water and 30% ethanol extracts in antioxidant properties and studied the preventive effect of oxidative stress-induced muscle cells death for the first time. The part of introduction was changed line 59~63.
2.In introduction, the author mention about the oxidation and sarcopenia, but in the conclusion, the mechanism of anti-sarcopenia for this materia was not showed.
=> In conclusion, we changed the sentence (line 254). We mentioned that the sarcopenia is not but muscle aging.
I look forward to hearing from you concerning the acceptability of our manuscript in the future.
With best regards.
Sincerely,
Dong-Seon Kim, Ph.D.
Chief Research Scientist
Encl.
Co-authors

Round 2
Reviewer 1 Report
The Authors slightly improved the quality of the manuscript, but many concept are still missing. I will recommend the paper for publication only after significant revision
Specific comments
Abstract
As chemical characterization to be added I meant that of Jakyakgamcho-tang, not of ABTS. I think it should be mentioned in the abstract as it is an important part of the study.
Introduction
Line 62 - 63: please rewrite the sentence, it is not fully understandable for the reader
Results
I ask the Authors again: a statistical comparison between JGT-W and JGT-E phytochemicals, total polyphenols content and antioxidants activity (Figure 3, Table 1 and Table 2) should be assessed to highlight significative differences to be reported in the Results section.
Figure 4B: the Authors did not answer to the question why cells treated with extracts and H2O2 (starting from extracts 100 ug/mL) showed an enhancement of cell viability with respect to the untreated controls. If the Authors show such outcome they should be able to discuss it, because it makes little sense from a scientific point of view
Discussion
The discussion has not been improved, the Authors just added some sentence here and there. In the discussion section the results should be properly discussed, explaining the importance of the obtained data even in comparison with what has already been found in other researches. As it is now it seems merely a bunch of sentences without rhyme or reason
Author Response
October 27, 2020
Molecules
Editor:
Ref.: Ms. ID. Molecules-960068
Thank you for email on October 20, 2020, informing us of your comments on our manuscript (Manuscript ID: molecules-960068, Title: Effect of Jakyakgamcho-tang Extracts on H2O2-Induced C2C12 Myoblasts). I would now like to resubmit the revised manuscript. The manuscript has been modified based on your comments and corrections, our replies to which are included below. In the manuscript, we mark the highlight in the changed part.
Response to Reviewer 1 Comments
Abstract
As chemical characterization to be added I meant that of Jakyakgamcho-tang, not of ABTS. I think it should be mentioned in the abstract as it is an important part of the study.
- Added (line 18~20).
Introduction
Line 62 - 63: please rewrite the sentence, it is not fully understandable for the reader
- Rewritten
Results
I ask the Authors again: a statistical comparison between JGT-W and JGT-E phytochemicals, total polyphenols content and antioxidants activity (Figure 3, Table 1 and Table 2) should be assessed to highlight significative differences to be reported in the Results section.
- We added the statistical significance results between JGT-W and JGT-E in Fig. 3, Table 1 and 2.
Figure 4B: the Authors did not answer to the question why cells treated with extracts and H2O2 (starting from extracts 100 ug/mL) showed an enhancement of cell viability with respect to the untreated controls. If the Authors show such outcome they should be able to discuss it, because it makes little sense from a scientific point of view
- Rewritten in discussion part.
Discussion
The discussion has not been improved, the Authors just added some sentence here and there. In the discussion section the results should be properly discussed, explaining the importance of the obtained data even in comparison with what has already been found in other researches. As it is now it seems merely a bunch of sentences without rhyme or reason
- Rewritten
I look forward to hearing from you concerning the acceptability of our manuscript in the future.
With best regards.
Sincerely,
Dong-Seon Kim, Ph.D.
Chief Research Scientist
